# Long Non-Coding RNA *74687* Regulates Meiotic Progression and Gonadal Development in Rainbow Trout (*Oncorhynchus mykiss*) via the miR-15a-5p–*ccne1* Regulatory Axis

**DOI:** 10.3390/ijms26168036

**Published:** 2025-08-20

**Authors:** Tianqing Huang, Baorui Cao, Enhui Liu, Wei Gu, Yunchao Sun, Kaibo Ge, Gaochao Wang, Datian Li, Peng Fan, Ruiyan Xing, Gefeng Xu

**Affiliations:** 1State Key Laboratory of Mariculture Biobreeding and Sustainable Goods, Heilongjiang River Fisheries Research Institute, Chinese Academy of Fishery Sciences, Harbin 150070, China; huangtianqing@hrfri.ac.cn (T.H.); c3012304124@163.com (B.C.); liuenhui@hrfri.ac.cn (E.L.); guwei@hrfri.ac.cn (W.G.); sunyunchao@hrfri.ac.cn (Y.S.); gekaibo@hrfri.ac.cn (K.G.); gaochaowang@ymail.com (G.W.); lidatian@hrfri.ac.cn (D.L.); fanpeng@hrfri.ac.cn (P.F.); 13059088876@163.com (R.X.); 2College of Animal Science and Technology, Northeast Agricultural University, Harbin 150030, China

**Keywords:** lncRNA, *ccne1*, miR-15a-5p, gonadal development, meiosis

## Abstract

High-throughput transcriptomic analyses have identified numerous candidate miRNA–mRNA and long non-coding RNA (lncRNA) regulatory networks in teleosts, but most remain without systematic functional validation or mechanistic definition. Here, by interrogating miRNA–lncRNA networks in rainbow trout (*Oncorhynchus mykiss*) gonads, we define their roles in meiotic progression and gonadal development. From preliminary screening, we identified *lncRNA74687* as a central node and characterised its function. Subcellular localisation showed predominant nuclear enrichment of *lncRNA74687* in gonadal cells. Dual-luciferase assays confirmed miR-15a-5p targeting of Cyclin E (*CCNE1*) and *lncRNA74687*. Functional studies showed that concurrent overexpression of *lncRNA74687* and inhibition of miR-15a-5p synergistically increased the CCNE1 protein to maximal levels. 5-ethynyl-2′-deoxyuridine (EdU) assays showed that knockdown of *lncRNA74687* and *CCNE1* in rainbow trout gonadal (RTG-2) cells reduced proliferation by 36.4% and 41.2%, respectively (*p* < 0.05). Immunofluorescence indicated that *lncRNA74687* increased Synaptonemal Complex Protein 1 (SYCP1) signalling 6.93-fold in gonadal cells via CCNE1. In vivo, *lncRNA74687* knockdown increased miR-15a-5p expression 6.34-fold relative to the wild-type controls (*p* < 0.01). Transcriptomic profiling revealed broad downregulation of meiosis-related genes in *lncRNA74687*-deficient gonads, with the strongest reduction in *mstrg1* expression, indicating a key role of *lncRNA74687* in germ-cell meiotic progression. Together, these data show that *lncRNA74687* enhances *CCNE1* mRNA and the CCNE1 protein in rainbow trout by competitively binding miR-15a-5p. This *lncRNA74687*–miR-15a-5p–*CCNE1* axis regulates gonadal cell proliferation and meiotic gene expression during gonadal development.

## 1. Introduction

Long non-coding RNAs (lncRNAs) are epigenetic regulatory molecules measuring longer than 200 nucleotides (nt) [1]. They are structurally similar to mRNAs but lack protein-coding capacity and show low sequence conservation [2]. As key regulators of gene expression, lncRNAs participate in essential biological processes such as cell proliferation, differentiation, metastasis, and apoptosis through diverse mechanisms [3,4]. Based on the relationship between their genomic loci and functional genes, lncRNAs are classified into six major categories: sense, antisense, bidirectional, enhancer, intronic, and long intergenic [5]. Mechanistically, some lncRNAs act as competitive endogenous RNAs (ceRNAs) that regulate mRNA expression by competing with mRNAs for miRNA binding, whereas others serve as miRNA precursors, contributing to miRNA biogenesis and regulating target mRNA expression [6,7].

Fertility and reproductive development are fundamental biological processes that ensure species continuity, with meiosis serving as a critical step in gamete formation in sexually reproducing organisms [8]. In fish, chromosomes become indistinct during the late pachytene stage of meiosis, making them invisible under microscopy [9,10]. Cyclin E (CCNE), a key factor in cell cycle regulation, is involved in assembling the chromatin replication pre-complex and plays essential roles in coordinating cell cycle progression, promoting cell growth, and maintaining normal cell cycle function [11]. Overexpression of CCNE1 induces chromosomal instability in mouse hepatocytes [12], and the CCNE1–CDK2 complex plays a critical role in ovarian development and oogenesis in the spotted shrimp *Penaeus monodon* [13]. Regulatory network studies on CCNE have expanded to the lncRNA–miRNA–mRNA axis. For example, the SENP3–EIF4A1–miR-195-5p axis in humans promotes carcinogenesis by regulating CCNE1 expression [14].

Rainbow trout (*Oncorhynchus mykiss*), an economically important cold-water species, is widely farmed in the Americas, Europe, and Asia [15]. It is also a widely used model organism in toxicology, nutrition, immunology, and reproductive physiology [16]. Our previous work identified an lncRNA–miRNA–mRNA network comprising miR-15a-5p, *ccne1*, and three lncRNAs (*lncRNA74687.3*, *lncRNA76205.1*, and *lncRNA30919.1*) that regulate meiosis and gonadal development, and play a key role in meiotic arrest [17].

To investigate the regulatory mechanism of key miRNA–lncRNA interaction networks in rainbow trout gonads and their role in meiosis and gonadal development, we used a dual-luciferase reporter assay to verify the targeting relationship between miR-15a-5p and *ccne1*, as well as the three lncRNAs. We then examined the regulatory functions of *lncRNA74687*, *ccne1*, and miR-15a-5p in meiosis and gonadal development at both the cellular and organismal levels using quantitative real-time PCR (qPCR), protein immunoblotting (Western blot), and immunofluorescence (IF). These findings provide a theoretical basis for understanding the molecular mechanisms of reproductive regulation in fish.

## 2. Results

### 2.1. Defining the Target Genes of miR-15a-5p

To elucidate the molecular mechanism of the miRNA–lncRNA interaction network regulating gonadal development in rainbow trout, the predicted binding sites of miR-15a-5p in the *ccne1* gene and three candidate lncRNAs (*lnc76205.1*, *lnc30919.1*, and *lnc74687.3*) were identified (Figure 1A). Wild-type and mutant dual-luciferase vectors (pmirGLO-ccne1-WT/MUT and pmirGLO-lncRNAs-WT/MUT) were constructed for *ccne1*, *lnc76205.1*, *lnc30919.1*, and *lnc74687.3*. Sequencing confirmed that all vector sequences matched the target sequences (Figure 1B).

To validate the interaction between miR-15a-5p and the predicted targets and to identify the lncRNAs involved in the miRNA–lncRNA interaction network, we co-transfected the dual-luciferase vectors with miR-15a-5p mimic, miR-15a-5p inhibitor, mimic NC, or inhibitor NC into 293T cells and measured the relative light units (RLU). The RLU of the miR-15a-5p mimic group was 1.16 times higher than that of the *ccne1* wild-type group (*p* < 0.05), and the RLU of the *ccne1* wild-type group was 1.27 times higher than that of the miR-15a-5p inhibitor group (*p* < 0.01) (Figure 1C). No significant effect of miR-15a-5p was observed on *lnc30919.1* or *lnc76205.1* (Figure 1D,E). For *lnc74687.3*, the RLU of the wild-type group was 1.26 times higher than that of the miR-15a-5p mimic group (*p* < 0.01), and the RLU of the miR-15a-5p inhibitor group was 1.23 times higher than that of the wild-type group (*p* < 0.05) (Figure 1F). In summary, miR-15a-5p inhibited bis-luciferase activity by targeting the 3′UTR of *ccne1* and *lnc74687.3*, identifying these two transcripts as target genes of miR-15a-5p.

### 2.2. Characterisation of lnc74687.3

Based on the 1648 nt full-length sequence of *lnc74687.3* from rainbow trout gonads obtained in our previous study [17], the minimum free energy (MFE) secondary structure was predicted using RNAfold (default parameters). The results showed that *lnc74687.3* formed a complex stem–loop structure (MFE = −410.70 kcal/mol), indicating a stable secondary structure (Figure 2A). Coding potential, a key criterion for identifying lncRNAs, was assessed using CPC2.0 Online website, which indicated that *lnc74687.3* has no coding capacity (Figure 2B). Sequence conservation analysis using the NCBI database retrieved no homologous sequences, suggesting that *lnc74687.3* is weakly conserved. Subcellular localisation of lncRNAs is closely related to their biological functions and potential molecular roles [18]. Therefore, nuclear and cytoplasmic RNAs were separately isolated from gonadal tissues. Using β-actin as an internal reference, RT-PCR was performed to verify the expression of the nuclear marker gene (*Gapdh*) and the cytoplasmic marker gene (*Spata*), as well as to assess the relative expression of *lnc74687.3*. The results showed that *lnc74687.3* is predominantly localised in the nucleus (Figure 2C,D). In conclusion, *lnc74687.3* is a long non-coding RNA predominantly expressed in the nucleus of rainbow trout gonocytes.

### 2.3. Co-Expression of miR-15a-5p, Ccne1, and lncRNA74687.3

To investigate the relationship between miR-15a-5p, *ccne1*, and *lncRNA74687.3*, different combinations of expression vectors (pcDNA3.1, p-*ccne1*, p-*lnc74687.3*) and regulatory molecules (siRNA, miRNA mimic/inhibitor, and their controls) were co-transfected into rainbow trout gonadal (RTG-2) cells, and gene expression was assessed by quantitative real-time PCR. Overexpression of *lnc74687.3* increased its transcript level by 28.49-fold, decreased miR-15a-5p expression to 0.26-fold, and increased ccne1 expression by 6.93-fold. Conversely, knockdown of *lnc74687.3* increased miR-15a-5p expression by 3.74-fold but did not significantly affect *ccne1* transcript levels. Co-overexpression of miR-15a-5p and *lnc74687.3* resulted in 12.54-fold and 2.91-fold increases in their respective expression levels. Overexpression of *ccne1* alone elevated *lnc74687.3* expression by 6.86-fold and reduced miR-15a-5p expression to 0.35-fold (Figure 3A–C). The effects of *lnc74687.3* and miR-15a-5p manipulation on CCNE1 protein expression were further examined by Western blot (Figure 3D). Knockdown of *lnc74687.3* reduced CCNE1 protein to 15% compared with 18% in the control, whereas co-overexpression of *lnc74687.3* and miR-15a-5p increased CCNE1 protein to 56%. These findings indicate that *lnc74687.3* may alleviate the inhibitory effect on *ccne1* and CCNE1 by competitively binding miR-15a-5p.

### 2.4. lnc74687.3 and Ccne1 Promote Gonocyte Proliferation in Rainbow Trout

The proliferative effects of *lnc74687.3* and *ccne1* on rainbow trout gonadal RTG-2 cells were assessed using 5-ethynyl-2′-deoxyuridine (EdU) assays. Overexpression of *lnc74687.3* increased the proliferation rate to 21.9%, significantly higher than that of the empty vector control group. In contrast, the proliferation rate in the siRNA-NC control group was 62.6%, whereas *lnc74687.3* knockdown significantly reduced it to 26.2% (Figure 4B). Overexpression of *ccne1* increased the proliferation rate to 25.6%, while in the siRNA-NC control group, it was 54.5%; knockdown of *ccne1* significantly reduced the rate to 21.4% (Figure 4C). In conclusion, *lnc74687.3* and *ccne1* promote gonadal cell proliferation.

### 2.5. lnc74687.3 Affects the Expression of Synaptonemal Complex Protein 1 (SYCP1)

Synaptonemal complexes (SCs) are associated with homologous chromosome synapsis [19], and SYCP1 and SYCP3 serve as markers for meiosis in fish [20]. Using cellular immunofluorescence, we examined the effects of overexpression or knockdown of *lnc74687.3* and *ccne1* on SYCP1 and SYCP3 protein expression. SYCP1 fluorescence intensity in groups overexpressing *ccne1* or *lnc74687.3* was significantly higher than in empty vector controls by 1.18-fold and 1.23-fold, respectively. In contrast, inhibition of *ccne1* or *lnc74687.3* significantly reduced SYCP1 fluorescence intensity in RTG-2 cells relative to the siRNA-NC controls, with the control intensities being 1.18-fold and 1.05 times higher, respectively (Figure 5A). However, neither overexpression nor knockdown of *lnc74687.3* or *ccne1* affected SYCP3 fluorescence intensity in RTG-2 cells (Figure 5B,C).

### 2.6. lnc74687.3 Regulates Ccne1 Gene Expression In Vivo by Targeting miR-15a-5p

Oocytes in the gonadal tissues of 1-year-old juvenile female rainbow trout are at the pre-minus I diplotene lineage stage or coarser. Double-stranded *lnc74687.3* RNA (ds-*lnc74687.3*) generated by in vitro transcription was administered via seven intraperitoneal injections over six consecutive weeks, with samples collected 72 h after each injection (Figure 6A). Quantitative PCR analysis of gonadal tissue showed that ds-*lnc74687.3* significantly reduced *lnc74687.3* expression by 1.63-fold, 6.71-fold, and 2.94-fold compared with the control group from weeks 2 to 4 post-injection (Figure 6B). *ccne1* mRNA levels were significantly reduced at weeks 1 and 2 post-injection, decreasing by 2.73-fold and 1.25-fold, respectively, but increased to 1.4 times the control level at week 5 (Figure 6C). Western blot analysis showed that knockdown of *lnc74687.3* significantly reduced CCNE1 protein levels in gonadal tissues, with CCNE1 expression in the control group being 2.03 times higher than in the knockdown group (Figure 6D,F). Knockdown of *lnc74687.3* also significantly increased miR-15a-5p expression in gonadal tissues (Figure 6E). In conclusion, at the organismal level, *lnc74687.3* regulates *ccne1* mRNA and CCNE1 protein expression in rainbow trout gonads by targeting miR-15a-5p.

To further verify whether *lnc74687.3* influences gonadal meiosis in rainbow trout, we examined the expression levels of the meiosis-related genes DNA meiotic recombinase 1 (*dmc1*), recombination 8 (*rec8*), spindlin family member (*spindlin*), *β-tubulin*, and MutL homolog 1 (*mlh1*) in gonadal tissues after somatic knockdown of *lnc74687.3*. Knockdown markedly reduced the expression of *mlh1*, *rec8*, *dmc1*, and *β-tubulin* to 0.04, 0.08, 0.22, and 0.26, respectively, relative to the controls, while *spindlin1* expression showed only a moderate decrease to 0.59 (Figure 6G).

## 3. Discussion

### 3.1. lnc74687.3 and Ccne1 Are Target Genes of miR-15a-5p

Research on fish fertility control has primarily focused on histology, cytology, and reproductive endocrinology. Key genes such as cytochrome P450 family 19 subfamily A polypeptide 1a (*cyp19a1a*), anti-Müllerian hormone (*amh*), forkhead box L2a (*foxl2a*), and doublesex and mab-3-related transcription factor 1 (*dmrt1*) play central roles in reproductive regulation [20,21,22]. Studies on the role of lncRNAs in fish reproduction remain limited, with existing work largely confined to preliminary screening and validation [23,24,25], and no in-depth studies have been conducted to systematically analyse the molecular mechanism of fish lncRNA. Our previous comparative transcriptome analysis of diploid and triploid rainbow trout gonads led to the first prediction and identification of miRNA–mRNA/lncRNA interaction networks potentially involved in gonadal development, including several key lncRNAs that may regulate gonadal cell proliferation and meiosis [17]. Building on these findings, the present study focused on verifying the biological functions of *ccne1* and *lnc74687.3* as miR-15a-5p target genes, and systematically elucidating the molecular mechanisms of the *ccne1*–*lnc74687.3*–miR-15a-5p regulatory network in rainbow trout reproductive development, through multi-dimensional experiments at both the cellular and organismal levels.

### 3.2. lnc74687.3 as a Nuclear Non-Coding RNA May Be Involved in Transcriptional Regulation

Although significant progress has been made in whole-genome and transcriptome sequencing and in the screening of lncRNAs in rainbow trout, experimental validation of lncRNA function remains limited. Classical molecular biology techniques still play an irreplaceable role in lncRNA functional verification [26]. In this study, *lnc74687.3*, a key regulator of reproductive development in rainbow trout, was found to have a unique secondary structure comprising three main domains: a multifurcated head, an intermediate trunk, and a re-furcated tail. It is localised in the nucleus of gonadal cells and is non-coding, suggesting a potential role in transcriptional regulation. The structure and subcellular localization of an lncRNA can reflect its expression site and coding potential. A subset of lncRNAs is distributed in the nucleus and chromatin, where they can interact with DNA, RNA, and proteins to regulate chromosome structure and function [19,27], or influence mRNA splicing, stability, and translation by modulating gene transcription [28].

### 3.3. lnc74687.3 Affects Germ-Cell Meiotic Progression by Regulating Cell Cycle- and Meiosis-Related Protein Expression in Gonadal Cells

In multicellular organisms, precise regulation of cell division and differentiation is essential for organ function [29,30]. Regulation of cell proliferation during gonadal development, from the early to mature stages, was found to be particularly critical, with the cell cycle regulator CCNE1 playing a central role in coordinating cell cycle progression and maintaining normal cell division [13]. Since both *lnc74687.3* and *ccne1* influenced gonadal cell proliferation, it was inferred that *lnc74687.3* may affect rainbow trout fertility by modulating gonadal cell proliferation. Abnormal gonadal development disrupts meiosis, resulting in impaired gamete production. The transverse filament protein SYCP1 is essential for meiotic recombination and SC assembly, both of which are key to meiosis [31,32]. Knockdown of *lnc74687.3* significantly reduced SYCP1 expression in RTG-2 cells, suggesting that *lnc74687.3* may influence germ-cell meiotic progression by regulating the expression of key meiotic proteins, thereby affecting gametogenesis.

### 3.4. lnc74687.3 Affects Gamete Production by Regulating the miR-15a-5p/ccne1/CCNE1 Pathway

The gonads of rainbow trout at one year of age are in an early developmental stage, during which primordial germ cells (PGCs) differentiate into oogonia and undergo an asynchronous transition from mitosis to meiosis [33]. Knockdown of *lnc74687.3* in rainbow trout resulted in an abnormal increase in *ccne1* expression at week 5, which may reflect a conserved mechanism for initiating and maintaining gene expression during development. As an important regulator of chromosome behaviour in meiosis, the *dmc1* gene encodes a recombinase essential for synapsis [8]. *Rec8* localises to the axial core of meiotic chromosomes, where it mediates cohesion between sister chromatids and is critical for homologous chromosome pairing [34,35]. As part of the DNA mismatch repair machinery, *mlh1* is essential for maintaining chromosome structure during the first meiotic division [36]. Abnormal meiosis in rainbow trout ovaries may also be linked to altered expression of *Spindlin-1* [37]. Additionally, the microtubule protein β-tubulin affects meiosis by influencing spindle formation, with its dimer structure serving as an essential spindle component [38]. Our previous research found that CCNE1 balance meiosis and gamete development through specific regulatory mechanisms, and their dysregulation may be a key factor underlying meiosis inhibition and reproductive development abnormalities [39]. In vivo validation confirmed our cellular findings: knockdown of *lnc74687.3* increased miR-15a-5p expression, while *ccne1* mRNA and CCNE1 protein were significantly reduced due to miR-15a-5p activity. Reduced expression of *dmc1*, *rec8*, *mlh1*, and β-tubulin correlated with a marked decrease in SYCP1 signal intensity in RTG-2 cells. Collectively, *lnc74687.3* regulates meiosis in gonadal cells via the miR-15a-5p/*ccne1*/CCNE1 pathway, thereby inhibiting cell proliferation and ultimately reducing gamete production (Figure 7).

## 4. Materials and Methods

### 4.1. Ethics Statement

All animal experiments were approved by the Ethics Committee for Animal Experiments of the Heilongjiang River Fisheries Research Institute, Chinese Academy of Fisheries Sciences (Approval No. 20250305-001), and conducted in strict compliance with the Guidelines for Ethical Review of Laboratory Animal Welfare in China.

### 4.2. Animals

The rainbow trout used in this study were supplied by the Bohai Sea Cold Water Fish Experimental Station (Mudanjiang), Heilongjiang River Fisheries Research Institute, Chinese Academy of Fisheries Sciences. All fish were acclimated for two weeks before the experiment. They were reared in a 1 m^3^ aquarium with a recirculating water system, with water temperature maintained at 10 ± 0.5 °C, dissolved oxygen at 7.8–10.0 mg/L, and pH at 7.2–7.5.

In vivo injection experiments were conducted using 48 one-year-old rainbow trout, divided into control and knockdown groups, housed separately in the same aquarium. Fish were anaesthetised with MS-222 (250 mg/L) before each injection to prevent pain. After the start of the experiment, three individuals from each group were randomly selected weekly for euthanasia and sampling. All euthanasia procedures involved MS-222 (250 mg/L) anaesthesia followed by bloodletting to ensure a painless death. Collected tissue samples were immediately frozen in liquid nitrogen and then stored at −80 °C.

### 4.3. Cell Culture

Rainbow trout gonadal cells (RTG-2) and human embryonic kidney epithelial cells (293T) were maintained in our laboratory. RTG-2 cells were cultured in Minimum Essential Medium (MEM, Sigma, St. Louis, MO, USA) supplemented with 10% foetal bovine serum (FBS, Zeta Life, Adams Drive, CA, USA), 100 U/mL penicillin, and 100 µg/mL streptomycin (Biosharp, Hefei, China) at 18 °C in a low-temperature incubator with 5% CO_2_ and 21% O_2_. 293T cells were cultured in Dulbecco’s Modified Eagle Medium (DMEM, Sigma, St. Louis, MO, USA) supplemented with 10% FBS (Zeta Life, Adams Drive, CA, USA), 100 U/mL penicillin, and 100 µg/mL streptomycin (NCM Biotech, Suzhou, China) at 37 °C in an incubator with 5% CO_2_ and 21% O_2_. RTG-2 and 293T cells reached 80–90% confluence after 48 h and 24 h, respectively, and were passaged using 0.25% trypsin (NCM Biotech, Suzhou, China).

### 4.4. Dual-Luciferase Reporter Assays

The potential binding sites of miR-15a-5p in the *ccne1* gene and lncRNAs (*lnc76205.1*, *lnc30919.1*, and *lnc74687.3*) were predicted using TargetScan (https://www.targetscan.org/vert_42/, accessed 15 December 2019). Based on these predictions, primers containing the binding sites were designed with XhoI and SalI restriction sites at both ends, and the amplified fragments were cloned into pmirGLO dual-luciferase reporter vectors to generate wild-type and mutant constructs for *ccne1* and each lncRNA. Small interfering RNAs (siRNAs) targeting miR-15a-5p, *ccne1*, *lnc76205.1*, *lnc30919.1*, and *lnc74687.3* were synthesised by Jimagene. At 48 h post-transfection, double luciferase activity was measured using the dual-luciferase reporter assay system (Promega, Madison, WI, USA), and relative fluorescence activity was calculated as the ratio of firefly luciferase to Renilla luciferase activity. Target genes of miR-15a-5p were confirmed by comparing the relative fluorescence activity changes across reporter constructs.

### 4.5. Subcellular Fractionation

Subcellular RNA isolation was performed under clean conditions using a Cytoplasmic and Nuclear Fractionation RNA Purification Kit (Norgen, Thorold, ON, Canada). Briefly, gonadal tissues frozen at −80 °C were ground in liquid nitrogen, homogenised in Lysis Buffer J, and centrifuged at 14,000× *g* for 10 min. The supernatant (cytoplasmic fraction) and pellet (nuclear fraction) were separated. Buffer SK and anhydrous ethanol were added to each fraction, combined, washed, and eluted through silica gel membrane columns to obtain high-purity cytoplasmic and nuclear RNA. The subcellular distribution of *lnc74687.3* was quantified by RT-qPCR.

### 4.6. RNA Extraction and Real-Time PCR Analysis

Total RNA from tissues was extracted using a Simply P Total RNA Extraction Kit (BioFlux, San Francisco, CA, USA) following the manufacturer’s instructions. cDNA was synthesised using the BioFlux reverse transcription kit (BioFlux, San Francisco, CA, USA). Gene expression was quantified by RT-qPCR using SYBR Green (NovaBio, Shanghai, China) according to the manufacturer’s protocol: 95 °C for 10 min; 39 cycles of 95 °C for 10 s and 60 °C for 30 s; and a melting curve from 65 °C to 95 °C with increments of 0.5 °C/s. Relative gene expression was calculated using the 2^−ΔΔCt^ method, with β-actin as the internal reference. Specific primers were designed using Primer Premier 5 (Table 1).

### 4.7. Western Blot Analysis

Cell or tissue extracts were mixed with 2× SDS-PAGE loading buffer (Beyotime Shanghai, China), boiled for 10 min, and separated by SDS-PAGE, followed by transfer to PVDF membranes (Biosharp, Hefei, China). After blocking for 1.5 h in tris-buffered saline with tween-20 (TBST) containing 5% BSA and 0.5% Triton X-100, the membranes were incubated with primary antibodies against CCNE1 (1:10,000 in TBST; ABclonal Technology, Wuhan, China) and β-actin (1:10,000 in TBST; Immunoway, Plano, TX, USA). Goat anti-rabbit IgG secondary antibody (1:2000 in TBST; Cell Signalling Technology, Danvers, MA, USA) was then applied. After each step, the membranes were washed five times with TBST. All incubation and washing solutions fully covered the PVDF membranes. Protein bands were visualised using an Odyssey imaging system (LI-COR Inc., Lincoln, NE, USA) and quantified with ImageJ.

### 4.8. Cell Proliferation Assay

Cell proliferation was assessed using an EdU Cell Proliferation Kit with Alexa Fluor 555 (Beyotime, Shanghai, China) following the manufacturer’s instructions with minor optimisation. Briefly, cells were incubated overnight with EdU working solution 48 h after transfection, fixed in 4% paraformaldehyde for 30 min at room temperature, permeabilised with 0.5% Triton X-100 in PBS for 15 min, and reacted with Click reaction solution (containing Alexa Fluor 555) for 30 min in the dark. Nuclei were counterstained with Hoechst (Beyotime, Shanghai, China) for 5 min. Between steps, cells were washed three times with PBS (5 min each). Images were captured under a fluorescence microscope. EdU-positive cells were quantified using ImageJ, and proliferation rates were calculated using Hoechst-stained nuclei as a reference.

### 4.9. Bioinformatics Analysis

Minimum free energy secondary structures were predicted using RNAfold (http://rna.tbi.univie.ac.at/, accessed 27 May 2024) with default parameters. The coding potential of lncRNAs was assessed using CPC2 software (http://cpc2.cbi.pku.edu.cn/, accessed 27 May 2024).

### 4.10. Statistical Analysis

All independent experiments included at least three parallel groups. Data were analysed and plotted using IBM SPSS Statistics v27.0 (IBM, Armonk, NY, USA) and GraphPad Prism v8.0.2 (GraphPad Software, San Diego, CA, USA). The results are presented as the mean ± standard deviation. Comparisons between two groups were performed using unpaired *t*-tests, and comparisons among multiple groups were performed using one-way ANOVA.

## 5. Conclusions

In summary, we report the first identified mechanism of lncRNA action in fish. In rainbow trout, *lnc74687.3* is a long non-coding RNA localised in the nucleus of gonadal cells. *lnc74687.3* regulates *ccne1* mRNA and CCNE1 protein expression by targeting miR-15a-5p, thereby influencing cell proliferation and meiosis. This lncRNA is a potential key factor in regulating meiosis and gonadal development in rainbow trout gonadal cells. The validation of this key lncRNA regulatory factor provides a target element for future in-depth research into rainbow trout reproductive capacity and lays crucial groundwork for understanding targeted regulatory mechanisms and advancing reproductive regulation technology.

## Figures and Tables

**Figure 1 ijms-26-08036-f001:**
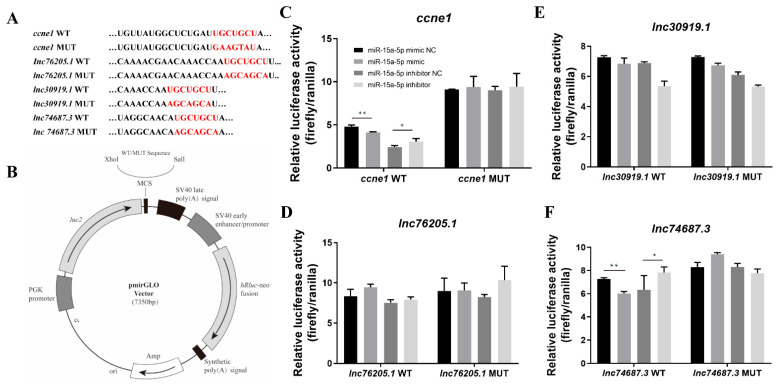
Construction of dual-luciferase vectors and analysis of relative light units. (**A**) Target sites in *ccne1*, *lnc76205.1*, *lnc30919.1*, and *lnc74687.3* wild-type and mutant sequences (red words). (**B**) Structure of pmirGLO-sequence-WT/MUT plasmid. (**C**–**F**) Relative fluorescence activity of *ccne1*, *lnc76205.1*, *lnc30919.1*, and *lnc74687.3* transfected into 293T cells. Data are expressed as mean ± SD (*n* = 3), with differences indicated by asterisks (** *p* < 0.01, * *p* < 0.05).

**Figure 2 ijms-26-08036-f002:**
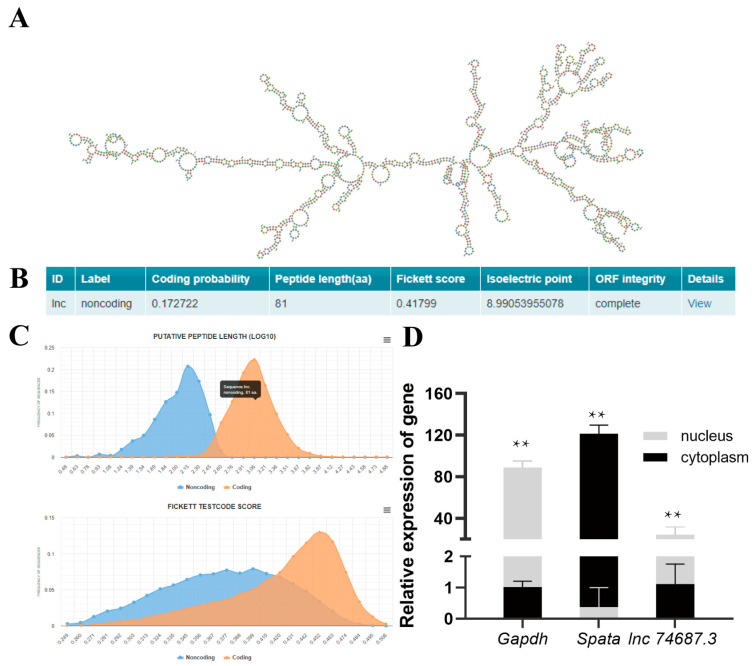
Structure and expression distribution of *lnc74687.3*. (**A**) Predicted secondary structure. (**B**,**C**) Coding capacity prediction. (**D**) Expression of *Gapdh*, *Spata*, and *lnc74687.3* in nucleus and cytoplasm of gonadal cells. Data are expressed as mean ± SD (*n* = 3), with differences indicated by asterisks (** *p* < 0.01).

**Figure 3 ijms-26-08036-f003:**
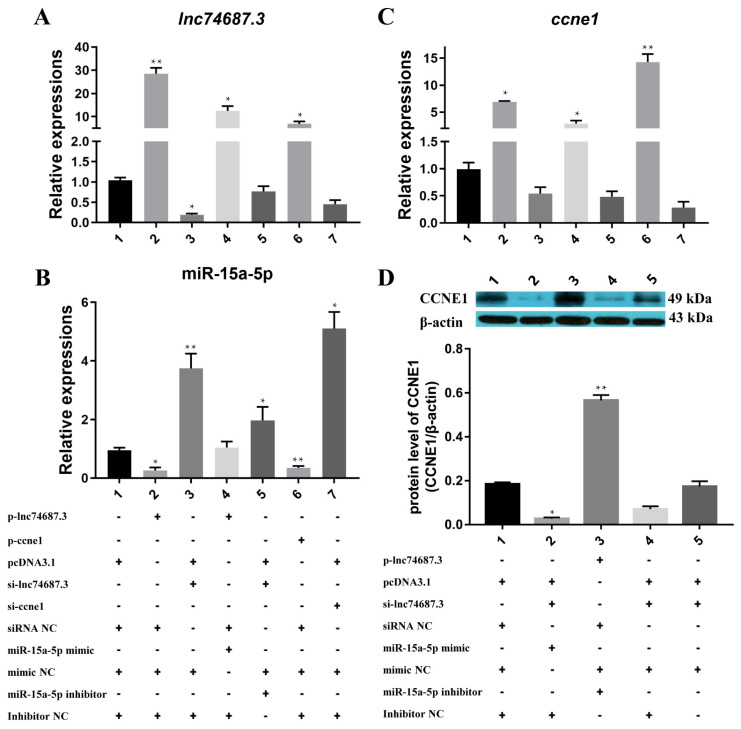
Co-expression analysis of miR-15a-5p, *ccne1*, and *lncRNA74687.3*. (**A**–**C**) Expression levels of *lnc74687.3*, *ccne1*, and miR-15a-5p under different combinations, with lower left table showing reagent combinations corresponding to horizontal axis (+, reagent added; -, reagent not added). (**D**) CCNE1 protein expression levels, with lower right table showing protein expression combinations (+, reagent added; -, reagent not added). Data are expressed as mean ± SD (*n* = 3), with differences indicated by asterisks (** *p* < 0.01, * *p* < 0.05).

**Figure 4 ijms-26-08036-f004:**
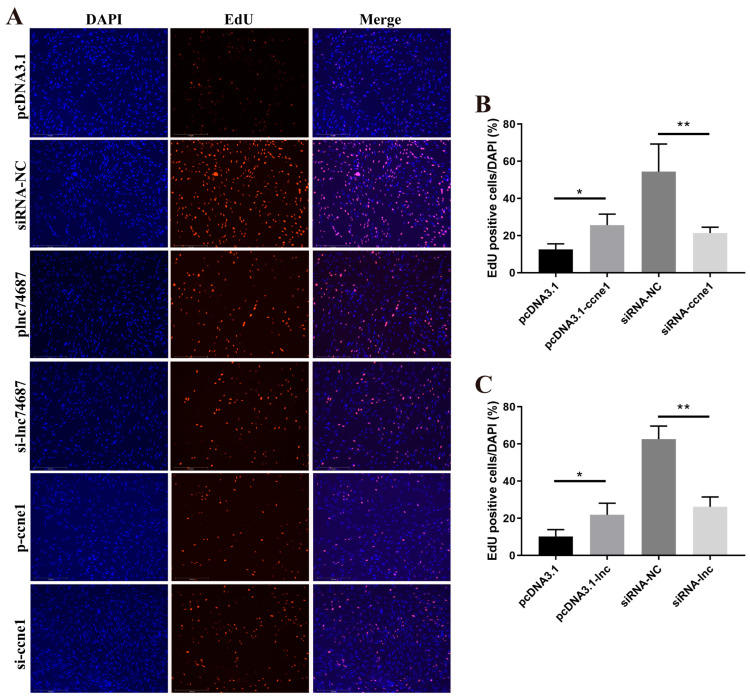
5-ethynyl-2′-deoxyuridine (EdU) assay of effect of *ccne1* on proliferative capacity of rainbow trout gonadal (RTG)-2 cells. (**A**) EdU results for RTG-2 cells under different transfection conditions. (**B**) Quantification of *lnc74687.3*’s effects on proliferation by ImageJ (ImageJ, 1.53q). (**C**) Quantification of *ccne1*’s effects on proliferation by ImageJ. Data are expressed as mean ± SD (*n* = 3), with differences indicated by asterisks (** *p* < 0.01, * *p* < 0.05).

**Figure 5 ijms-26-08036-f005:**
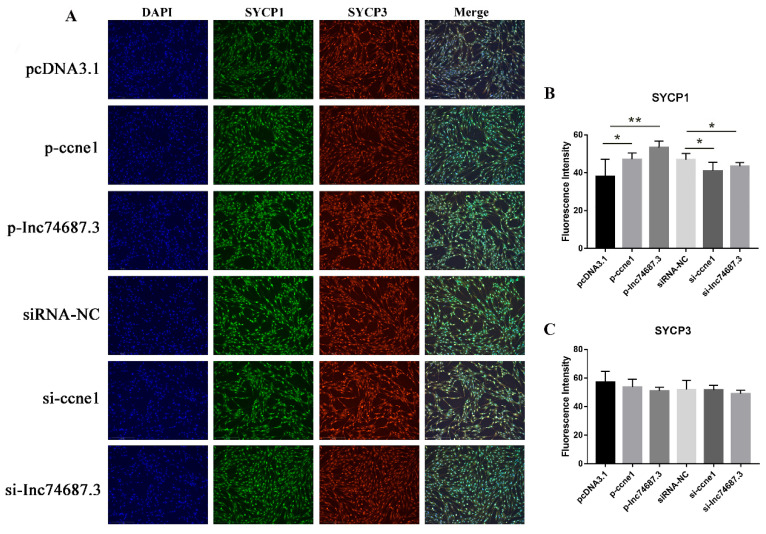
Cellular immunofluorescence detection of CCNE1, and *lnc74687.3*’s effects on expression of Synaptonemal Complex Protein 1 (SYCP1) and Synaptonemal Complex Protein 3 (SYCP3). (**A**) Immunofluorescence images of SYCP1 and SYCP3 in RTG-2 cells under different transfection conditions. (**B**) Quantification of SYCP1 fluorescence intensity by ImageJ. (**C**) Quantification of SYCP3 fluorescence intensity by ImageJ. Data are expressed as mean ± SD (*n* = 3), with differences indicated by asterisks (** *p* < 0.01, * *p* < 0.05).

**Figure 6 ijms-26-08036-f006:**
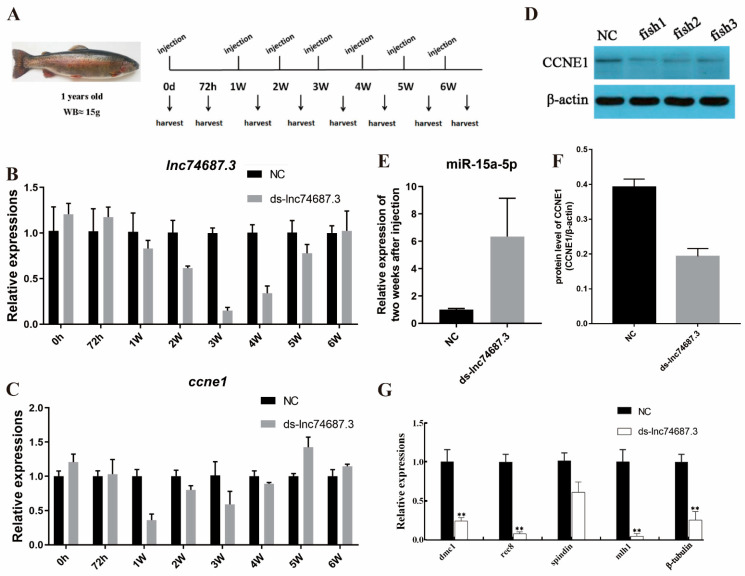
*lnc74687.3* targeting miR-15a-5p at the organismal level affects *ccne1* and CCNE1 expression. (**A**) Injection and sampling scheme; (**B**) *lnc74687.3* expression in rainbow trout gonads at different time points; (**C**) *ccne1* expression in rainbow trout gonads at different time points; (**D**) CCNE1 protein expression levels at weeks 1, 2, and 5; (**E**) miR-15a-5p expression two weeks after *lnc74687.3* knockdown; (**F**) CCNE1 protein expression levels; (**G**) effect of *lnc74687.3* knockdown on meiosis-related genes *dmc1*, *rec8*, *spindlin*, *β-tubulin*, and *mlh1*. Data are expressed as mean ± SD (*n* = 3), with differences indicated by asterisks (** *p* < 0.01).

**Figure 7 ijms-26-08036-f007:**
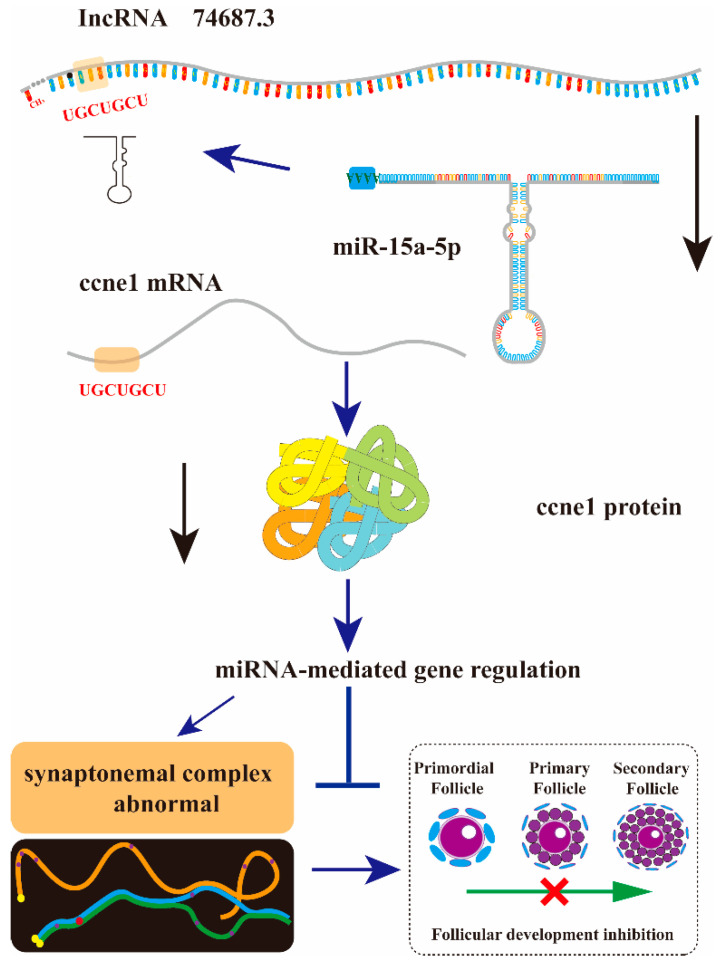
Mechanistic map of the *lnc74687.3*/miR-15a-5p/*ccne1* axis regulating gonocyte meiosis and gonadal development. *lnc74687.3* in rainbow trout gonads affects *ccne1* and CCNE1 expression by targeting miR-15a-5p, ultimately influencing gonadal cell meiosis and gonadal development.

**Table 1 ijms-26-08036-t001:** Oligonucleotide primers used for RT-qPCR.

Primer Name	Nucleotide Sequence (5′-3′)	Accession Number
dmc1-F	GCACGAGCGTCAAACATTCC	XM 021575970.2
dmc1-R	CGACCTCATCCTCCACAACT	
rec8b-F	GCAGTATGGGGTGGTTAT	XM 021562658.2
rec8b-R	CTCGTCATGGAATTGGTC	
mlh1-F	CAGGATTGTGGAGGTGGTGA	XM 021606940.2
mlh1-R	ATGGGTGAGAGTTCTTGGGA	
β-tubulin-F	CGACCCCACTGGCACATA	XM 021601756.2
β-tubulin-R	CCTCACCACATCCAAAAC	
spindling-F	CCATACCAGGCGGTGATACA	XM 021588921.2
spindling-R	CAACCACAGGAGGCTGAAGA	
ccne1-F	AACGGCAACGTCTGATTTTC	LOC 110486705
ccne1-R	CGTGGGATATAATGCCAAGC	
gapdh-F	CATTGAGGGTCTGATGAGCA	NM 001124209.1
gapdh -R	CCTCCACAGCTTTCCAGAAG	
spata4-F	ATTTTGTAAGCGTGTGTG	NM 001124526.2
spata4-R	TTTGGTTGGAAGTGATGT	
lnc74687.3-F	CAGGCCTCCATTCCAAATTA	MSTRG. 74687
lnc74687.3-R	ACGTGAGTTGAACACGCAAC	
β-actin-F	CTCACCGACTACCTGATGAAGATC	LOC 100136352
β-actin-R	GTAGCACAGCTTCTCCTTGATGTC	
miR-15a-5p	TAGCAGCACAGAATGGTTTGTGA	

## Data Availability

All data generated or analysed during this study are included in this article.

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
