# Peer review of "Long Non-Coding RNA 74687 Regulates Meiotic Progression and Gonadal Development in Rainbow Trout (Oncorhynchus mykiss) via the miR-15a-5p–ccne1 Regulatory Axis"

_ijms, 2025, doi:10.3390/ijms26168036_

Round 1
Reviewer 1 Report
Comments and Suggestions for Authors
The study addresses an interesting topic regarding the role of lnc74687.3 in regulating ccne1 expression via miR-15a-5p in rainbow trout, both in vitro and in vivo. While the experimental design is promising and the biological question relevant, there are several issues that must be addressed to improve the clarity, scientific rigor, and presentation of the manuscript. Below are my detailed comments:
The abstract is too long and lacks a clear structure.
Gene names should be italicized throughout the manuscript, following standard scientific conventions.
Species names should be italicized as per nomenclature rules throughout the manuscript.
There are numerous typos and syntax issues throughout the manuscript that make it difficult to read, particularly in the Results section.
Lines 62–66: This section lacks references. Please provide relevant citations to support the statements made.
Lines 71–74: The meaning of the sentence is unclear due to poor syntax. Please rewrite for clarity and scientific accuracy.
The aim of the study should be clearly and explicitly stated at the beginning of the final paragraph of the Introduction section.
Figure 1: The sub-panels C, D, E, and F are presented without a proper caption or explanation. These must be added to make the figure interpretable.
Line 159: The sentence “ccne1 was activated alone” is vague and confusing. Please explain what is meant — was ccne1 overexpressed?
In the EdU-based proliferation assay (Figure 4), the siRNA-NC control group showed a proliferation rate of 62.6%, much higher than that of lnc74687.3 overexpression (21.9%). This is counterintuitive, as negative controls should not show higher activity than overexpression conditions. The same inconsistency appears for ccne1. Please revisit the text and clarify the control groups and data interpretation. The current explanation is unclear due to syntax issues.
Please clarify why SYCP1 and SYCP3 proteins were selected for analysis.
Subsection 2.6: The title includes “in vitro”, but the experiment described is clearly in vivo (whole animal injections). Please correct the title.
In this same subsection, there is inconsistent reporting of fold changes. Please ensure all fold-changes are clearly and consistently reported.
The increase in ccne1 mRNA at week 5 after knockdown is not discussed or explained. This unexpected rebound requires interpretation.
Subsection 2.7 is very short and lacks sufficient explanation. Please elaborate on the results and their biological meaning.
Lines 278–279: The suggestion that this ceRNA network may facilitate cancer treatment is not substantiated. Please justify this claim using current literature or remove it.
The Discussion section is generally poor and needs substantial improvement. The findings should be interpreted in the context of existing literature, compared with similar studies and framed within a broader biological and translational context. Please also include a paragraph discussing the strengths and limitations of your study, as well as suggestions for future research directions.
The number of fish used in in vivo experiments is not reported, nor are the biological replicates per group. Please add this essential information to the Methods section.
Comments on the Quality of English LanguageThe manuscript requires extensive English language editing. Numerous sections, particularly in the Results, contain grammatical errors, awkward phrasing, and unclear sentence structure, which make the scientific content difficult to follow. In several cases, the meaning of key sentences is ambiguous due to syntax issues.
I strongly recommend that the authors have the manuscript reviewed by a native English speaker to improve clarity, flow, and precision.
Reviewer 2 Report
Comments and Suggestions for Authors
- Right now the manuscript explore many molecular interactions, but the main hypothesis and the specific aims are not so well delimited. It would help a lot to rewrite them at the end of the introduction, in a very explicit way, so the reader can really understand what is being tested and the biological relevance of the lnc74687.3/miR-15a-5p/ccne1 pathway.
-
The study show effect on meiosis in rainbow trout, but the link with reproductive physiology or aquaculture is a bit weak. Adding one or two paragraphs connecting the molecular findings with practical or ecological implications would give more value to the work.
-
The functional validation miss some important controls. For example, including a CCNE1 overexpression without lnc74687.3 could show if the effect is really because of the interaction network or only the protein itself. Also, a non-related lncRNA as negative control would help to discard non-specific effects.
-
In Materials and Methods some key info is missing: exact number of fish used per experimental group, number of biological replicates for each assay, and any inclusion/exclusion criteria. These details are essential for reproducibility and proper evaluation of the statistics.
-
The abstract is very long and include too many experimental details. Making it more structured (Background & Objectives, Methods, Results, Conclusion) and focusing only on the key findings would make it easier to read and more attractive to the audience.
Round 2
Reviewer 2 Report
Comments and Suggestions for Authors
The authors incorporated all my feedback. I have nothing further to add.